# Experimental and Quantum Chemical Investigations on the Anticorrosion Efficiency of a Nicotinehydrazide Derivative for Mild Steel in HCl

**DOI:** 10.3390/molecules27196254

**Published:** 2022-09-23

**Authors:** Nadia Betti, Ahmed A. Al-Amiery, Waleed Khalid Al-Azzawi

**Affiliations:** 1Materials Engineering Department, University of Technology-Iraq, Baghdad 10001, Iraq; 2Department of Chemical and Process Engineering, Faculty of Engineering and Built Environment, Universiti Kebangsaan Malaysia (UKM), Bangi 43000, Malaysia; 3Energy and Renewable Energies Technology Center, University of Technology-Iraq, Baghdad 10001, Iraq; 4Department of Medical Instruments Engineering Techniques, Al-Farahidi University, Baghdad 10001, Iraq

**Keywords:** nicotinehydrazide, mild steel, corrosion inhibitor, DFT, EIS

## Abstract

A corrosion inhibitor namely N′-(4-hydroxy-3-methoxybenzylidene) nicotinohydrazide was synthesized and the inhibition efficiency of the investigated inhibitor toward the mild steel corrosion in 1 M HCl was studied. The anticorrosion effect has been investigated by weight loss (WL) techniques and electrochemical analysis includes potentiodynamic polarization (PDP) studies and electrochemical impedance spectroscopy (EIS). The current investigation has demonstrated that the tested inhibitor is suitable in corrosive environment and the inhibitive efficacy up to 97% in 1 M HCl. PDP measurements showed that the nicotinohydrazide is a mixed type inhibitor. EIS measurements showed that an increase in the inhibitory concentration leads to an increase in the charge transfer resistance ($R_{ct}$) and a decrease in the double-layer capacitance ($C_{dl}$). Experimental results for the inhibitory performance of WL methods and electrochemical techniques (PDP and EIS) are in good agreement. The tested inhibitor molecules adsorbed on the surface of mild steel in a hydrochloric acid solution followed Langmuir’s isothermal adsorption. Quantum chemical parameters based on density function theory (DFT) techniques were conducted on oxygen/nitrogen-bearing heterocyclic molecule employed as a corrosion inhibitor for mild steel in HCl to evaluate the correlation between the inhibitor structure and inhibitory performance. The parameters including the energy gap (Δ*E*), dipole moment (*μ*), electronegativity (*χ*), electron affinity (*A*), global hardness (*η*), softness (*σ*), ionization potential (*I*), the fraction of electrons transferred (Δ*N*), the highest occupied molecular orbital energy (EHOMO), and the lowest unoccupied molecular orbital energy (ELUMO) were also calculated and were in good agreement with the experimental results.

## 1. Introduction

Due to its unique physical properties, mild steel is widely applied in industrial applications. Unfortunately, due to the harsh conditions they are in, corrosion is sometimes inevitable. To avoid catastrophes caused by steel corrosion, Li and colleagues [1] previously called for corrosion data to be shared. Metallic materials are commonly employed in a variety of applications, such as construction, particularly in the petroleum, oil, and gas industries [2], but most metals are thermodynamically unstable in their pure form, and thus are easily corroded. Corrosion is a major source of safety problems, as well as financial loss, equating to a global loss according to the National Association of Corrosion Engineers (NACE) of US$2.5 trillion, around 3.5% of global GDP. Corrosion costs can be either direct or indirect [3,4,5]. Direct costs comprise the expense of repairing, storing, and replacing damaged metallic equipment, as well as the cost of converting alloys to metals and vice versa, for example, the cost of nickel plating and galvanization [6,7]. Moreover, the direct cost of corrosion includes economic losses related to the synthesis, characterization, and application of compounds as corrosion inhibitors. Corrosion has indirect expenses, such as leakage of liquids (petroleum) and gases from transport pipelines, which negatively impact machine performance and transportation efficiency. Rust and scale contamination (corrosion products) can reduce the number of materials that can be conveyed, and impair transport efficiency and machine performance by clogging valves and couplings. Furthermore, the leakage of transported petroleum-based liquids and gases is linked to a variety of environmental issues because of their toxicity, thus corrosion scientists and engineers have devised several corrosion mitigation techniques. Inorganic compounds, such as nitrites, chromates, borates, molybdates, silicates, and zinc salts, were first used to mitigate corrosion (before 1960) [8] by producing a highly effective passive coating over metal surfaces. Between the 1960s and 1980s, they were largely superseded by more cost-effective alternatives including phosphonic acid, gluconates, polyacrylates, surface-active chelates, polyphosphates, polyphosphonates, phosphonates, and carboxylates [8]. Since these substances precipitate at the metal-environment contact, they are referred to as precipitating inhibitors. Subsequently, between 1980 and 1995, harmful chemicals were gradually replaced with natural alternatives, such as natural biopolymers, biosurfactants, vitamins, tannins, and natural compounds. More recently, rare earth metals (REM), polyfunctional compounds, the synergism of organic/inorganic compounds employing REM, and the encapsulation of inhibitors have been key areas of research (1995 to present) [9]. These options have very minimal or no toxicity while providing excellent protection and because of their efficiency, economy, ecology, and environmental friendliness (E4), organic chemicals have become one of the most successful and profitable means of corrosion inhibition [10,11,12]. The most efficient and cost-effective corrosion protection technique is the use of natural and/or synthetic organic molecules as corrosion inhibitors [1,13]. These inhibitors typically contain acetylenic bonds and/or aromatic rings in their molecular structures as well as heterocyclic atoms such as phosphorous, sulfur, oxygen and nitrogen. According to the majority of researchers, organic inhibitors protect the metal by adhering to the substrate surface and creating a protective layer [1,14]. Thus, contemporary corrosion science and engineering research efforts are focused on developing corrosion inhibitors with hydrophilic polar functional substituents in their molecular structures.

This study evaluated the effectiveness of a mild steel corrosion inhibitor since mild steel is the most widely used building material and is also used to build fuel systems that come into contact with fuels containing various concentrations of corrosive solution. The nicotinehydrazide derivative “N′-(4-hydroxy-3-methoxybenzylidene) nicotinohydrazide” was created and tested as a corrosion inhibitor (Figure 1) as it is easy to synthesize and is of low cost. Scanning electron microscopy and weight loss techniques revealed that the inhibitor was effective in corrosive media, with an inhibitory efficacy of up to 97%.

## 2. Results and Discussion

### 2.1. Weight Loss Investigations

Figure 2 presents a summary of the weight loss experimental findings for mild steel specimens in a test corrosive environment with/without the addition of nicotinehydrazide derivative. The results showed that the nicotinehydrazide derivative protects the surface from corrosion and that the anticorrosion performance improved with nicotinehydrazide derivative concentration. The corrosion rate decreased with increasing inhibitor concentration at 303 K for 5 h immersion. The greatest inhibition efficiency (95.8%) was observed at 0.5 mM nicotinehydrazide derivative. The strong inhibition performance is attributed to the binding of nicotinehydrazide derivative molecules to the surface of the mild steel, which is supported by the presence of many heterogeneous atoms (three nitrogen atoms and three oxygen atoms) in the large molecular structure of the nicotinehydrazide derivative. Moreover, the type of substituent group connected to nicotinehydrazide significantly affects the corrosion inhibition ability. The addition of a phenyl group improves the electron density at the active sites, increasing the interaction between the inhibitor and the mild steel surface [15].

The ability of the tested inhibitor to inhibit corrosion increases with increasing concentration up to 0.5 mM because the inhibitor molecules are adsorbed onto the surface of the mild steel, creating a protective barrier. The adsorption of the inhibitor molecules from the surface of the mild steel causes the inhibitory efficiency to remain nearly constant when the inhibitor concentration exceeds 0.5 mM and reaches 1.0 mM.

### 2.2. The Effect of Exposure Time

Mild steel was exposed to 1 M hydrochloric acid in the presence of the inhibitor (0.1 to 1.0 mM) for 1 h to 48 h at 303 K to investigate the effect of exposure time on the corrosion inhibition efficiency of the nicotine hydroxide derivative (Figure 3). With the increase of exposure up to 10 h, the inhibition efficacy increases rapidly, slowly decreasing thereafter from 10 to 24 h, then decreases faster from 24 to 48 h. The increase in nicotine hydrazide derivatives (due to an increase in concentration) adsorbed on the surface of mild steel with increased exposure time increases the inhibition efficiency. However, as more inhibitor molecules are adsorbed on the mild steel surface, the adsorption density of the inhibitor increases dramatically, allowing for bonding between inhibitor molecules and iron atoms on the mild steel surface via van der Waals force (physical adsorption) and coordination bonds (chemical adsorption). Some inhibitor molecules may leave the surface, thus reducing the effective area covered by the inhibitor and reducing the inhibitory efficacy. The stability of the inhibitor layer adsorbed in the presence of 1 M HCl solution is evidenced by the relatively high inhibition efficiency observed during the long immersion period.

### 2.3. The Effect of Temperature

After 5 h of immersion at different temperatures (303–333 K), the corrosion inhibition of mild steel in 1 M HCl in the presence of different concentrations of the tested inhibitor (0.1–1.0 mM) was examined using the mass reduction technique. The rate of corrosion increased with temperature at the same inhibitor concentration (Figure 4) and anticorrosion efficacy decreased with increasing temperature from 303 to 333 K. The tested inhibitor was most effective at normal temperature. Physical adsorption is suggested by a decrease in inhibitory effectiveness with increased temperature increase at all concentrations. Additionally, the surface of mild steel loses inhibitor molecules at high temperatures due to desorption.

### 2.4. Adsorption Isotherm

Understanding the interaction between the inhibitor molecules and the metal surface is made easier by the adsorption isotherm. The surface coverage (*θ*) values for the nicotinehydrazide derivative, which were collected by weight loss methods, were used to assess which isotherms best suit the data. The inhibitor molecules attach to the surface of the metal either physically or chemically, so various adsorption isotherms (Temkin, Freundlich, and Langmuir isotherms) were used to analyze the adsorption process. The regression coefficient (R2) for the nicotinehydrazide derivative of 0.99713 suggested that the Langmuir adsorption isotherms fit the data very well, with the calculated slope and intercept values for the Langmuir isotherm of 1.06301 ± 0.04035 and −0.09668 ± 0.02238 respectively. Figure 5 depicts the Langmuir adsorption isotherm plot between *C*/*θ* and *C*, and the equation (Equation (1)) is as follows:(1)$$\frac{C}{\theta }=\frac{1}{K_{ads}}+C$$
where  C is the inhibitor concentration, θ represents the surface area, and $K_{ads}$ is the constant of equilibrium. 

In order to determine the free energy $\Delta G_{ads}^{o}$ of adsorption, a linear straight fitted plot between C/θ and C was used to determine the $K_{ads}$ value.

The following equation (Equation (2)) relates $K_{ads}\;$ with $\Delta G_{ads}^{o}$.
(2)$$\Delta G_{ads}^{o}=-RT\;ln\left(55.5K_{ads}\right)$$
where 55.5 refers to the water concentration (M), R is the constant of gas and T represents the temperature. 

The values for $\Delta G_{ads}^{o}$ were derived by including $K_{ads}$ in the aforementioned calculation.

Chemical adsorption is indicated by values of $\Delta G_{ads}^{o}$ ranging between $-40\;kJmol^{-1}$ and greater negative values, whereas electrostatic interaction, or physical adsorption, is indicated by values of $\Delta G_{ads}^{o}$ about or less negative than $-20\;kJmol^{-1}$ [16,17]. The $\Delta G_{ads}^{o}$ value for the nicotinehydrazide derivative of $-35.57\;kJmol^{-1}$ suggests that there are two different types of adsorption, chemical and physical.

### 2.5. Potentiodynamic Polarisation Measurements

The technique is demonstrated in Figure 6 and Equation (3) illustrates how the inhibitive efficacy was evaluated:(3)$$IE\left(\%\right)=\frac{i_{corr}-i_{corr\left(inh\right)}}{i_{corr}}\times 100$$
where $i_{corr\left(inh\right)}$ and $i_{corr}\;$ represent the current density in the presence/absence of the nicotinehydrazide derivative respectively.

Figure 7 exhibits the polarization curves for the mild steel samples in 1.0 M HCl solution in the absence and presence of various concentrations of nicotinehydrazide derivative at 303 K. Table 1 presents the experimental findings for the corrosion current density ($i_{corr}$), corrosion potential ($E_{corr}$), and inhibition efficiencies, with the anodic Tafel slope ($\beta_{a}$) and cathodic Tafel slope ($\beta_{c}$). The $\mathrm{Gamry}\;-{\;E}_{\mathrm{chem}}\mathrm{Analyzer}$ programme presents the Tafel fit strategy, which uses a nonlinear chi-square minimization to fit the data to the Stern–Geary equation.

The corrosion inhibitor can be classified as either a cathodic or anodic when the $E_{corr}$ shift reaches 85 mV [18,19]. Since the nicotinehydrazide derivative displaces most $E_{corr}\;$, the molecules can be regarded as mixed-type. The addition of the nicotinehydrazide derivative to the acidic medium slows down the anodic dissolution of mild steel and delays the cathodic hydrogen evolution. Table 1 demonstrates that the presence of the nicotinehydrazide derivative resulted in a decrease in $i_{\mathrm{corr}}$ values, therefore the corrosion rate decreased as the concentration of nicotinehydrazide derivative increased, improving the inhibitory efficacy. The values of the Tafel constants ($\beta_{a}$, $\beta_{c}$) barely changed in the presence of the nicotinehydrazide derivative, demonstrating that the nicotinehydrazide derivative was in charge of both processes and the adsorbed molecules had no effect on the hydrogen evolution or dissolution of mild steel [20].

### 2.6. Electrochemical Measurements

Electrochemical impedance spectroscopy (EIS) was used to determine the inhibition efficiency of the tested inhibitor in inhibiting corrosion. The experimental results for mild steel corrosion at 303 K, in the absence and presence of an inhibitor, are shown in Table 2 and Figure 8 shows the Nyquist plots. The addition of nicotinehydrazide derivative significantly increased the overall resistivity of mild steel in the HCl solution. Two loops were observed in the Nyquist plots: one loop in the high-frequency band (HF) and one loop at an intermediate frequency (MF), with less inductive effect at lower frequencies (LF). The HF and MF loops are attributed to the EIS instrument limits at high frequencies with low resistance and charge-transfer processes, accordingly. Accordingly, the inductive behavior observed in the LF region is caused by the relaxing process of the corrosion product adsorption or the inhibitor molecule adsorption onto the test surface in HCl in the absence/presence of the inhibitor, respectively [21]. 

The inhibition efficiency (*IE* %) was calculated from the charge-transfer impedance using Equation (4):(4)$$IE\left(\%\right)=\frac{R_{ct}^{\prime }-R_{ct}}{R_{ct}^{\prime }}\times 100$$
where $R_{ct}^{\prime }$ and $R_{ct\;}$ are the charge-transfer-resistance in the presence and absence of the nicotinehydrazide derivative.

Table 2 shows that the charge-transfer resistance ($R_{ct}$) increased as the inhibitor concentration increased, gradually corroding the systems due to large charge-transfer resistance [22]. Furthermore, increased inhibitor impedance is associated with decreased mild steel capacitance. The observed rise in the $C_{dl}$, which was related to an increase in the local dielectric constant and/or the thickness of the electrical double layer, shows that the nicotinehydrazide derivative adsorbed onto the material interface with the solution [23].

Nyquist diagrams reveal incomplete capacitive rings, as shown in Figure 8. The higher the ring radius, the more corrosion resistance is actually revealed. The width expands when nicotine hydrazide is added, indicating that it has a corrosion inhibitory action on mild steel in the test solution. This is caused by the inhibitor molecules adsorbing and forming a protective layer on the surface of the mild steel. The lower estimated capacitance of the $C_{dl}\;$ double layer indicates that the corrosion attack was reduced due to the film’s ability to cover the surface [24].

The observed rise in the $C_{dl}$ value in an acidic medium with the addition of nicotinehydrazide derivative may be attributed to inhibitor adsorption onto the most active adsorption centers [25]. The corrosion process decreased the homogeneity of the adsorbed nicotinehydrazide derivative layer.

Furthermore, the results showed that the IE% increased with increasing inhibitor concentration, following the same pattern as the inhibition efficiency determined by the potentiodynamic and mass-loss techniques. Figure 9 depicts the equivalent circuit diagram for equating the EIS findings for HCl solution both in the absence and presence of the inhibitor.

$R_{s}$ (solution resistance), a $CPE_{dl}\;\left(\mathrm{constant}\;-\;\mathrm{phase}\;\mathrm{element}\right)$, and $R_{ct}$ (charge-transfer resistance) are the circuit components and the charge-transfer resistance rating was used to determine how well the electrons travelled over the contact [26].

In Figure 9, $R_{s}$ is the resistance of the environment; $R_{ct\;}$ is the charge-transfer impedance; $CPE_{dl}\;$ is the double-layer constant phase.

A CNLS (complex nonlinear least squares) simulation was employed [26,27,28,29,30] since an equivalent circuit was used to calculate the simulated values and for comparison with experimental data.

### 2.7. Surface Morphology

Severe corrosion and damage were observed on the mild steel surface immersed for 5 h at 303 K in 1 M HCl medium (Figure 10a), whereas the addition of 0.5 mM nicotinehydrazide derivative protected the mild steel surface as evidenced by the smooth surface with fewer holes (Figure 10b).

### 2.8. Theoretical Calculation

Quantum chemistry investigations are used to effectively match inhibitor molecule corrosive inhibitive efficacy with predicted molecular orbitals (MOs) energy levels without the need for laboratory tests, saving time and money [31]. The energies of the frontier MOs (EHOMO and ELUMO), the separation energy (ELUMO EHOMO), *E* representing the function of reactivity, electron affinity, ionization potential, chemical softness (*σ*), chemical hardness (η), and electronegativity (*χ*), the number of transferred electrons (Δ*N*), and the dipole moment are computed quantum chemical variables to predict the protection performance of the test inhibitor (Table 3).

HOMO energy (EHOMO) refers to the ability of a molecule to donate a lone pair of electrons and the higher the EHOMO, the higher the ability to donate electrons to electrophilic molecules [32], whereas a molecule with a low ELUMO has more ability to accept electrons from metals. The differences in energy between EHOMO and ELUMO (that is, Δ*E*) inform the interaction of specific molecules and the smaller the difference, the greater the interaction, as shown in Table 3. The DFT findings demonstrated that the test inhibitor molecules have a small energy gap value compared to other reported inhibitors, hence, the highest reactivity [33]. 

The dipole moment gives information on the polarity and the higher the dipole moment, the higher the polarity [34]. Regarding the HOMO values in Table 3, the test inhibitor which has the highest inhibitory efficiency also has the highest EHOMO as computed using the DFT/B3LYP/6-31G basis set, confirming the experimentally determined ranking. According to the DFT/B3LYP/6-31G computations, the test inhibitor has a low LUMO in the aqueous phase, thus, it has the potential to interact with the mild steel surface.

The variation in energy between the HOMO and the LUMO (Figure 11) is significant for determining chemical reactivity, kinetic stability, chemical hardness/softness, and optical polarizability of organic compounds. Large values of Δ*E* result in strong electrical stability and minimal reactivity, whereas high values of Δ*E* imply high reactivity since electrons are easily excited and transferred from the HOMO to the LUMO. Lower Δ*E* values indicate a good organic corrosion inhibitor. According to the DFT/B3LYP/6-31G calculations, the tested inhibitor has the lowest value of Δ*E*, indicating that it has the strongest response. Regarding the decreasing values of Δ*N*, which is an electron transfer from the tested inhibitor to the mild steel surface, the efficiency of the tested inhibitor increases. The quantum parameters energy of HOMO (*E_HOMO_*) and LUMO (*E_LUMO_*), energy gap (Δ*EL–H*), electronegativity (*χ*), hardness (*η*), softness (σ), and the number of electrons transferred (Δ*N*) are listed in Table 3.

### 2.9. Mulliken Atomic Charges

Mulliken charges are frequently applied to measure atomic charges in the molecule and to identify inhibitor adsorption centers. Moreover, the ability of a heteroatom to adsorb onto a metallic substrate via a donor-acceptor interaction increases with its negative charge. Table 4 presents the atomic charges of the inhibitor molecules showing that the two nitrogen and oxygen atoms have high atomic charges (O(13) = −0.3420 and N(10) = −0.3113), suggesting that they are responsible for iron absorption. 

### 2.10. Mechanism of Inhibition

The adsorption of the inhibitor molecules onto the metallic surface is influenced by the inhibitor’s molecular structure, charge, how the acidic environment behaves, and the surface properties of the metallic surface. By inhibiting the active sites on the metal that are sensitive to corrosion, inhibitors allow metal to be absorbed from aqueous solutions. The inhibitory resistance of organic molecules is caused by the formation of a protective barrier that is adsorbed onto the metallic substrate. Weight loss measurements and electrochemical techniques revealed that the inhibitor considerably reduced mild steel corrosion. Additionally, the adsorption isotherm analyses suggested that the inhibitor molecules adhere to the mild steel surface in a manner that closely resembles that predicted by the Langmuir adsorption model. How the protective coating adheres to the mild steel surface is affected by: (1) electrostatic interactions with protonated heteroatoms, and (2) various links between inhibitor molecules [35]. The examined inhibitor molecule contains several heteroatoms and pi-bonds in addition to aromatic rings with lone pairs of electrons, which contribute to the creation of coordination bonds and significant adsorption onto the mild steel surface. The interaction between iron d-orbitals and the tested inhibitor molecules mostly followed chemical adsorption as indicated by the free energy parameter. The presence of heteroatoms in the nicotinohydrazide having electron pairs has promoted chemisorption onto the mild steel surface. Figure 12 presents the proposed mechanism of the corrosion inhibition of mild steel in a corrosive medium. 

## 3. Materials and Methods

### 3.1. Materials

The Company of Metal Samples provided the mild steel and the elemental composition is listed in Table 5. The mild steel samples were prepared according to ASTM G1-03 [36] and the surfaces were abraded using silicon carbide paper. All chemicals were obtained from Sigma-Aldrich (Selangor, Malaysia) and used without further purification. The acidic solution of 1 M hydrochloric acid was prepared by diluting concentrated HCl (37%) with distilled water.

### 3.2. Weight Loss Techniques

The mild steel samples were immersed in 100 mL of 1 M hydrochloric acid with/without the inhibitor (0.1, 0.2, 0.3, 0.4, 0.5 and 1.0 mM) as described in NACE TM0169/G31 [37] and placed in a water bath at 303, 313, 323, or 333 K for 1, 5, 10, 24, and 48 h. The samples were then removed and treated as specified in ASTM standard G1-03. Following the calculation, the average weight loss was used to obtain the corrosion rate [36]. The corrosion rate (*C_R_*), *IE* and surface coverage (θ) were determined according to Equations (5)–(7):(5)$$C_{R\;}\left(mg\cdot cm^{-2}\cdot h^{-1}\right)=\frac{W}{at}$$
(6)$$IE\%=\left[1-\frac{C_{R\left(i\right)}}{C_{R}{}_{o}}\right]\times 100$$
(7)$$\theta =1-\frac{C_{R\left(i\right)}}{C_{R}{}_{o}}$$
where W signifies the weight loss of the mild steel sample (mg), *a* is the surface area of the mild steel sample (cm^2^), *t* is the exposure time (h), $w_{o}$ is the weight loss of mild steel sample in 1 M HCl in the absence of the test inhibitor, and $w_{i}$ is the weight loss of the mild steel sample in 1 M HCl in the presence of the test inhibitor.

### 3.3. Computations

The quantum chemical computations were performed using Gaussian 09 [38]. The inhibitor structure in the gas phase was optimized via the B3LYP approach and the basis set of $6-31G^{++}\left(d,p\right)$. 

Based on Koopman’s theorem [39], the ionization potential (I) corresponds to $E_{HOMO}$, whereas electron affinity (A) corresponds to $E_{LOMO}$. Both (I) and (A) were calculated as per Equations (8) and (9): (8)$$I=-E_{HOMO}$$
(9)$$A=-E_{LOMO}$$

The electronegativity (χ), hardness (η), and softness (σ) can be calculated according to Equations (10)–(12):(10)$$\chi =\frac{I+A}{2}$$
(11)$$\eta =\frac{I-A}{2}$$
(12)$$\sigma =\eta^{-1}$$

The transferred electrons fractional number, ΔN, can be determined based on Equation (13) [39]
(13)$$\Delta N=\frac{\chi_{Fe}-\chi_{inh}}{2\left(\eta_{Fe}+\eta_{inh}\right)}$$

Here, $\chi_{Fe}$ and $\chi_{inh}$ are the electronegativities of the iron and tested inhibitor, whereas $\eta_{Fe}\;$ and $\eta_{inh}$ are the hardness of iron and tested inhibitor, respectively.

For mild steel (Fe), the value of Δ*N* can be determined according to Equation (14), were $\chi_{Fe}$ = 7 eV, $\eta_{Fe}$ = 0 eV:(14)$$\Delta N=\frac{7-\chi_{inh}}{2\left(\eta_{inh}\right)}$$

### 3.4. Electrochemical Data

Mild steel samples were cleaned according to ASTM G1-03 [36] and served as the working electrode throughout this study. The active mild steel sample area was 4.5 cm^2^ and the tests were performed at inhibitory doses of 0.1 to 0.5 M in 1.0 M hydrochloric acid solution that was ventilated but not stirred at 303 K. All measurements were performed in triplicate to calculate the mean on a Gamry Instrument Potentiostat/Galvanostat/ZRA type REF 600 using Gamry’s DC105 and EIS300 software. The dynamic current potential was changed from 0.25 to +0.25 V SCE at a scan rate of 0.5 mVs^−1^. All impedance values were matched to the appropriate equivalent circuits using the Gamry Echem Analyst tool (ECs). The inhibition of corrosion was assessed using a Gamry water-jacketed glass cell with three electrodes: the working electrode, the counter electrode, and the reference electrode and a saturated calomel electrode served as the reference electrode (SCE). Electrochemical measurements were initiated approximately 30 min after the working electrode was exposed to the corrosive environment to maintain the steady-state potential [40,41]. 

### 3.5. Surface Scanning Electron Microscope

The corrosive behavior of the acidic solution (1 M HCl) on the mild steel surface after 5 h of treatment without and with the addition of 0.0005 M PMBMH was assessed by a scanning electron microscope (Zeiss MERLIN Compact FESEM at the UKM Electron Microscopy Unit).

## 4. Conclusions

Nicotinehydrazide shows strong corrosion protection for mild steel in 1 M HCl due to the presence of highly effective electronic adsorption centers (O, N, and pi-bonds) that inhibit the active sites of the metal. The primary conclusions are as follows:The synthesized nicotinehydrazide derivative shows good inhibition efficiency for the mild steel corrosion in 1 M HCl environment and the inhibition efficiency increases on increasing the concentration of nicotinehydrazide and decreases with the increase in temperature. The highest inhibition efficiency was 97% at 303 K in 1 M HCl solution.Nicotinehydrazide participates in chemical adsorption on metallic surfaces and weakly bonds to the metal surface with the inhibition efficacy decreasing as the temperature increases.Nicotinehydrazide prevents mild steel corrosion due to the creation of a protective layer of inhibitor molecules at the steel–electrolyte interface.The G_ads_ suggests a chemisorption and physisorption phenomena, and the adsorption mechanism is spontaneous.The quantum chemical simulations indicate that nicotinehydrazide uses oxygen and nitrogen to adsorb onto a mild steel surface.There was good agreement between the experimental results and the theoretical analysis.

## Figures and Tables

**Figure 1 molecules-27-06254-f001:**
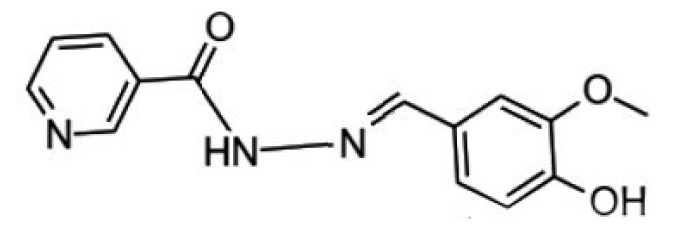
The molecular structure of nicotinehydrazide.

**Figure 2 molecules-27-06254-f002:**
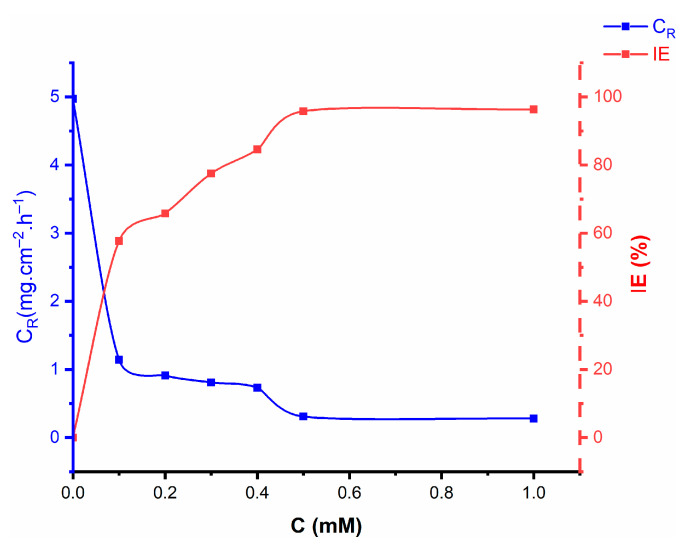
Effect of various concentrations of nicotinehydrazide derivative on the corrosion rate and inhibition efficiency for mild steel immersed in 1 M HCl for 5 h at 303 K.

**Figure 3 molecules-27-06254-f003:**
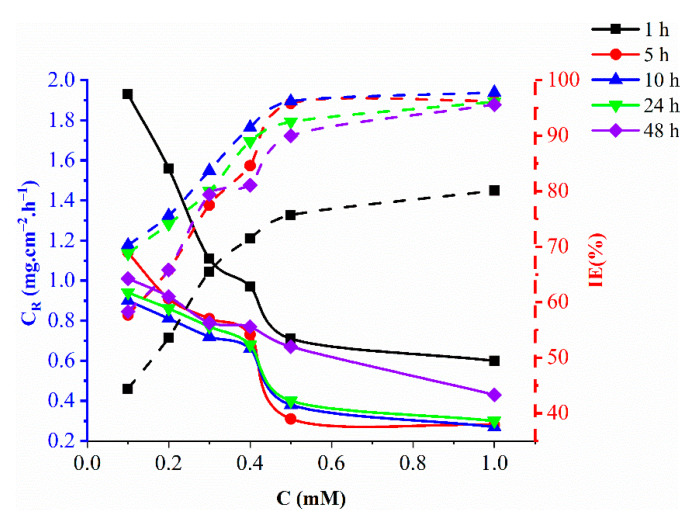
Effect of various concentrations of nicotinehydrazide derivative on the corrosion rate and inhibition efficiency of mild steel immersed in 1 M HCl for different times at 303 K.

**Figure 4 molecules-27-06254-f004:**
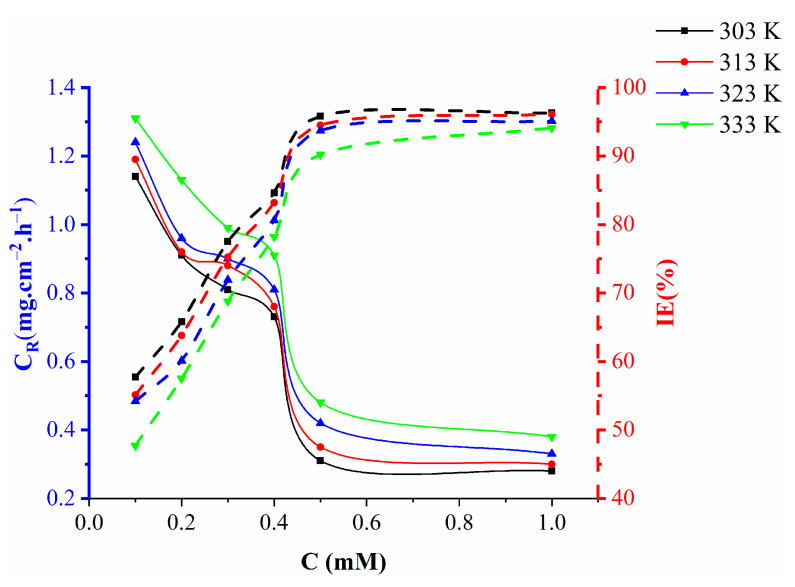
Effect of various concentrations of nicotinehydrazide derivative on the corrosion rate and inhibition efficiency of mild steel immersed in 1 M HCl at different temperatures for 5 h.

**Figure 5 molecules-27-06254-f005:**
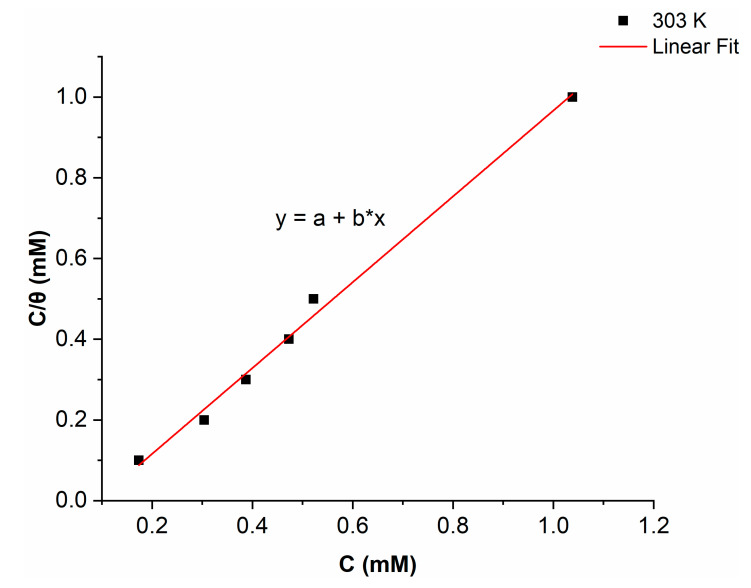
Langmuir adsorption isotherm for mild steel with the addition of the nicotinehydrazide derivative.

**Figure 6 molecules-27-06254-f006:**
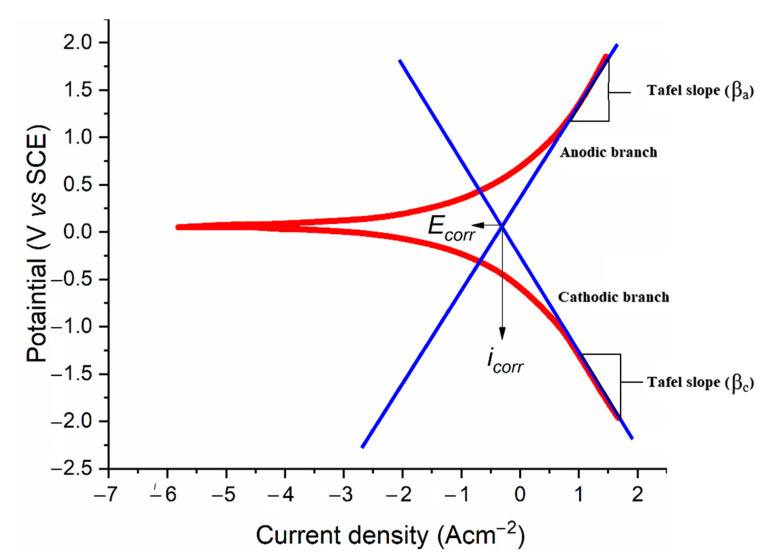
The Tafel slopes, corrosion potential, and corrosion current density were calculated via extrapolation.

**Figure 7 molecules-27-06254-f007:**
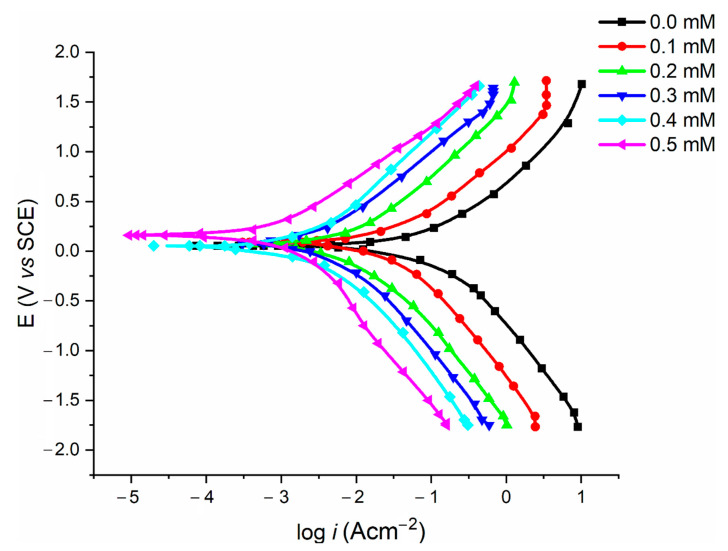
Polarization curves of mild steel samples in acidic media with various concentrations of nicotinehydrazide derivative.

**Figure 8 molecules-27-06254-f008:**
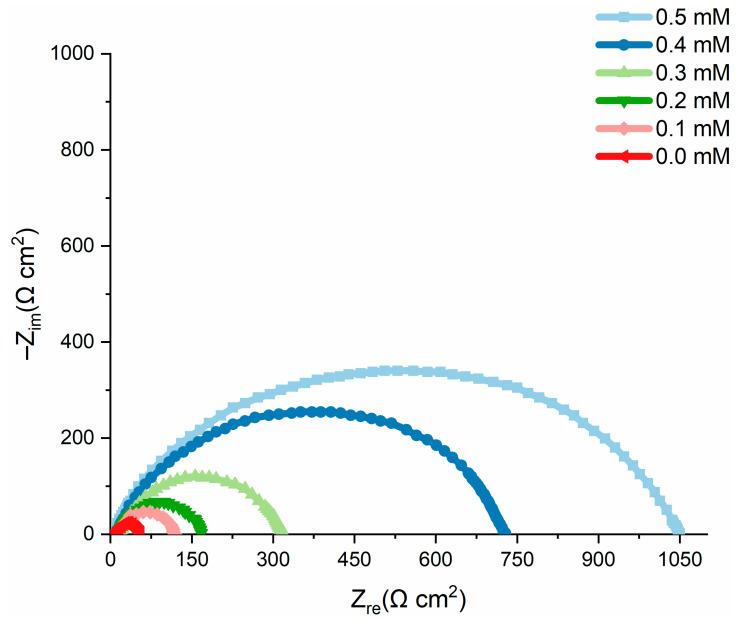
Nyquist plots of the test samples in acidic solution in the absence and presence of different concentrations of nicotinehydrazide derivative at 303 K.

**Figure 9 molecules-27-06254-f009:**
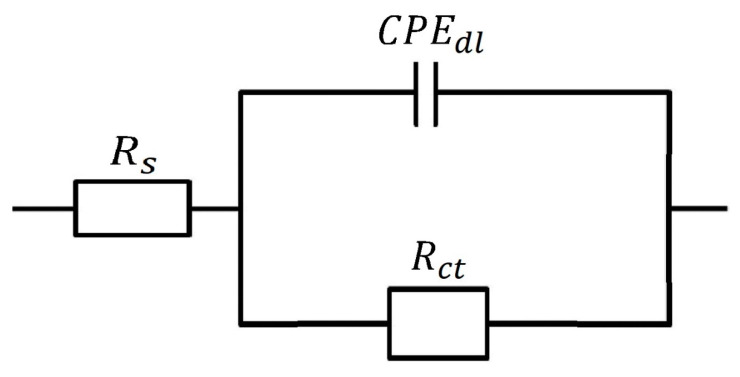
The equivalent circuit model was used to fit the experimental data.

**Figure 10 molecules-27-06254-f010:**
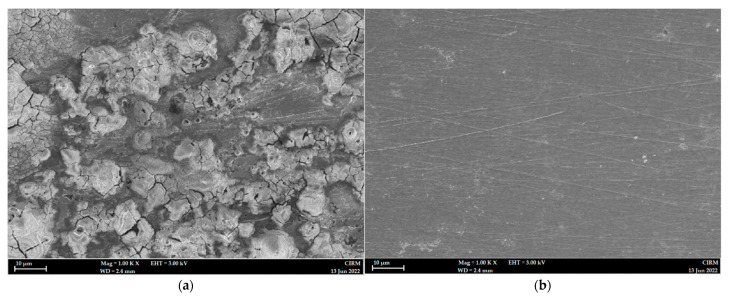
SEM photographs of mild steel samples in the absence (**a**), and presence of 0.5 mM nicotinehydrazide derivative (**b**) in 1 M HCl at 303 K.

**Figure 11 molecules-27-06254-f011:**
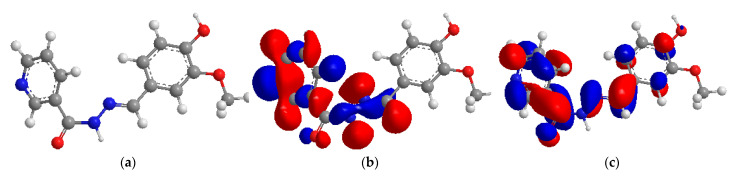
The optimized structure (**a**), HOMO (**b**) and LUMO (**c**) of the tested corrosion inhibitor.

**Figure 12 molecules-27-06254-f012:**
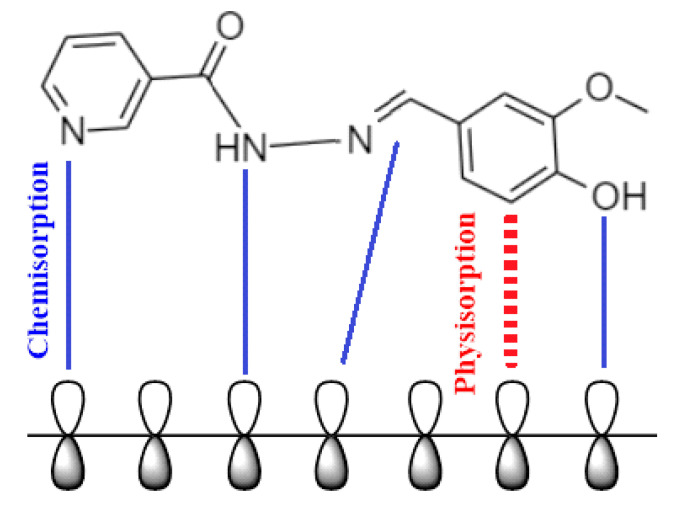
The proposed mechanism of corrosion inhibition of mild steel in 1 M HCl.

**Table 1 molecules-27-06254-t001:** Tafel parameters for mild steel samples in acidic media with and without various concentrations of nicotinehydrazide derivative.

*Conc.* mM	*E_corr_* (V)	*β_a_* (mV/dec)	*β_c_* (mV/dec)	*i_corr_* (μA·cm^−2^)	*IE* (%)
0.0	–0.47	240	220	515.3 ± 1.83	0
0.1	–0.51	125.7	188.6	355.7 ± 5.03	73.6
0.2	–0.49	91.8	151.8	199.3 ± 3.70	82.8
0.3	–0.55	83.4	131.2	95.7 ± 2.93	86.4
0.4	–0.53	56.7	126.7	84.8 ± 1.84	92.8
0.5	–0.46	48.9	102.8	58.3 ± 4.77	97.1

**Table 2 molecules-27-06254-t002:** EIS parameters for steel sample in absence and presence of various concentrations of nicotinehydrazide derivative in 1 M HCl at 303 K.

*Conc.* (mM)	*R_s_* (Ω cm^2^)	*R_ct_* (Ω cm^2^)	*C_dl_* (μF)	*IE* %
0.0	2.4	50.64	530	0
0.1	2.1	78.65	310	55.2
0.2	2.3	172.37	311	72.8
0.3	2.2	280.76	244	84.8
0.4	2.4	327.33	180	92.5
0.5	2.3	466.13	130	96.8

**Table 3 molecules-27-06254-t003:** Quantum chemical parameters for inhibitors at B3LYP/6-311G (d,*p*).

*E_HOMO_* (eV)	*E_LUMO_* (eV)	Δ*E* (eV)	*I* (eV)	*A* (eV)	*χ* (eV)	*η* (eV)	*σ* (eV^−1^)	*µ*	Δ*N*
−9.884	−3.922	−5.962	9.884	3.922	6.903	2.981	0.335	−1.455	0.145

**Table 4 molecules-27-06254-t004:** Mulliken atomic charges for the corrosion inhibitor.

**Atom**	**Charge**	**Atom**	**Charge**	**Atom**	**Charge**	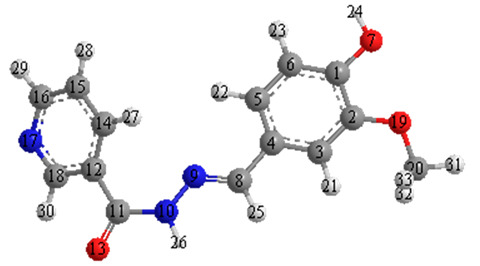
C(1)	0.0726	C(8)	−0.1199	C(15)	−0.1996	
C(2)	0.0539	N(9)	−0.0346	C(16)	−0.0467	
C(3)	0.1627	N(10)	−0.3113	N(17)	−0.1446	
C(4)	−0.0450	C(11)	0.3789	C(18)	−0.0192	
C(5)	−0.0874	C(12)	−0.1612	O(19)	−0.1862	
C(6)	−0.1985	O(13)	−0.3420	C(20)	−0.0797	
O(7)	−0.2252	C(14)	−0.0327	H(21)	0.1407	

**Table 5 molecules-27-06254-t005:** Mild steel chemical composition (wt%).

C	Mn	Si	Al	S	P	Fe
0.21%	0.05%	0.38%	0.01%	0.05%	0.09%	balance

## Data Availability

Not applicable.

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
