# Peer review of "Experimental and Quantum Chemical Investigations on the Anticorrosion Efficiency of a Nicotinehydrazide Derivative for Mild Steel in HCl"

_molecules, 2022, doi:10.3390/molecules27196254_

Round 1

Reviewer 1 Report

Betti et al. used experimental and theoretical calculation methods to study the corrosion inhibition performance of nicotinehydrazide derivative on mild steel in hydrochloric acid solution. There are large number of organic and inorganic corrosion inhibitors while some are toxic while others relatively less. A glance of corrosion science literature reveal that finding a corrosion inhibitor was never a challenge as evidenced by huge number of publications getting published every year. This paper could be one more addition to such data literature and I think could be consider after a major revision is carried out. My specific comments are as follows:

1. The language of the manuscript needs to be greatly improved.

2. The abstract is too cumbersome and should be simplified to a certain extent, and specific parameters of quantitative experimental results should be given.

3. The introduction part lacks a certain appeal, and the authors are advised to refer to Journal of Colloid and Interface Science 506 (2017) 478-485 and Colloids and Surfaces A: Physicochemical and Engineering Aspects 645 (2022) 128892.

4. The resolution of the molecular structure of nicotinehydrazide in Figure 2 is too low.

5. The Nyquist plots in Figure 9 should give the fitted values as well as the original values of the experiment, and some important frequency values should be marked.

6. Electrochemical impedance spectroscopy results should be discussed in more detail, and authors are advised to refer to Journal of Colloid and Interface Science 609 (2022) 838–851.

7. The corrosion inhibition mechanism of Nicotinehydrazide on steel should be discussed separately in the manuscript, and a diagram of the inhibition mechanism should be added.

8. Carefully check the format of the reference to ensure that it meets the requirements of the journal.

Author Response

Dear reviewer,

Thank you for your useful comments and sggestions

All were conducted pointby point, so please see the revised manuscript

Thank you

Best regards

Reviewer 2 Report

Betti et al. examine a Nicotinehydrazide Derivative Reduces the Corrosion of Mild 2 Steel in 1 M HCl: Experimental and Computational 3 Investigations. The article is written nicely, explaining all the aspects related to the study. It will help researchers who are working in the same field. I recommend its acceptance after a major revision followed by the editorial correction.

1.       Title should be modified as it does not give the scientific meaning of the research.

2.       Abstract should contain key findings of the work, please add it.

3.       Figure 1 is not properly visible, please update it.

4.       Fitted curves are missing from figure 9.

5.       The conclusion can be improved.

6.       Please exchange old papers with newly published papers.

https://doi.org/10.3390/lubricants10070157; https://doi.org/10.1016/j.molstruc.2022.133424;  https://doi.org/10.3390/lubricants10030043

7.       Please check the entire manuscript and remove grammatical errors.

Author Response

(The authors gave the same response as above.)

Round 2

Reviewer 1 Report

I have read the revised version of the Manuscript and found that the authors have taken into accounts the concerns that I raised. Thus I recommend it for publication.

Reviewer 2 Report

Agree